# *Wolbachia* Natural Infection of Mosquitoes in French Guiana: Prevalence, Distribution, and Genotyping

**DOI:** 10.3390/microorganisms12101994

**Published:** 2024-09-30

**Authors:** Emmanuelle Clervil, Amandine Guidez, Stanislas Talaga, Romuald Carinci, Pascal Gaborit, Anne Lavergne, Sourakhata Tirera, Jean-Bernard Duchemin

**Affiliations:** 1Unité d’Entomologie Médicale, Institut Pasteur de la Guyane, Cayenne 97300, French Guiana; eclervil@pasteur-cayenne.fr (E.C.);; 2Laboratoire d’Interaction Hôte-Virus, Institut Pasteur de la Guyane, Cayenne 97300, French Guiana

**Keywords:** *Wolbachia*, mosquitoes, French Guiana

## Abstract

*Wolbachia* are the most spread bacterial endosymbionts in the world. These bacteria can manipulate host reproduction or block virus transmission in mosquitoes. For this reason, *Wolbachia*-based strategies for vector control are seriously considered or have already been applied in several countries around the world. In South America, *Wolbachia* have been studied in human pathogen vectors such as sand flies and mosquitoes. In French Guiana, the diversity and distribution of *Wolbachia* are not well known in mosquitoes. In this study, we screened for *Wolbachia* natural infection in mosquitoes in French Guiana by using 16S rRNA, *Wolbachia* surface protein (WSP), and multi-locus sequence typing (MLST) molecular assays. A total of 29 out of 44 (65.9%) mosquito species were positive for natural *Wolbachia* infection according to the PCR results, and two *Wolbachia* strains co-infected three specimens of *Mansonia titillans*. Then, we analyzed the phylogenetic relationships among the *Wolbachia* detected. All of the tested specimens of *Aedes aegypti*, the major dengue vector of French Guiana, were negative. These results regarding *Wolbachia* strain, distribution, and prevalence in mosquitoes from French Guiana highlight *Wolbachia*–mosquito associations and pave the way for a future *Wolbachia*-based strategy for vector control in this Amazonian territory.

## 1. Introduction

*Wolbachia* bacteria conquered the insect world millions of years ago [1,2,3]. Their diversity and their biological and physiological impacts on their hosts have been well studied. *Wolbachia* were discovered in 1924 [4], and knowledge has been accumulating for a century. Meta-analyses suggest that more than 52% of all terrestrial arthropods are naturally infected by *Wolbachia*, making these bacteria the most common animal endosymbiont on Earth [5,6,7]. 

*Wolbachia* have started to be well studied in mosquitoes over the last decade in South America, especially in Brazil [8]. French Guiana, which is more than 90% covered by rainforest [9], offers an important source of biodiversity [10]. The French territory harbors nearly 250 mosquito species [11], with some of them known as vectors or having the potential to be vectors of pathogens that are important public health concerns. Surprisingly, almost nothing is known about the diversity of *Wolbachia* or the diversity of mosquitoes naturally infected by these bacteria.

In French Guiana, the presence of *Wolbachia* that naturally infect insects has already been shown in two species of termites [12] and in the native electric ant *Wasmannia auropunctata* [13]. On the South American continent, *Wolbachia* have also been detected in one species of turtle ant [14], in sand flies [15], and in mosquitoes such as *Culex quinquefasciatus* [16] and *Aedes fluviatilis* [17]. Its presence has also already been shown in filaria (*Mansonella* sp. and *Brugia* sp.) originating from red howler monkeys and *Dirofilaria immitis*, causing canine heartworm disease [18]. Globally, *Wolbachia* are classified into 17 different supergroups [19,20]. Depending on the strain of *Wolbachia*, functional interactions between the bacteria and its host may look like symbiosis, reproductive parasitism, facultative mutualism, or obligate mutualism [5,19]. Reproductive parasitism [21] means that every reproductive alteration process represents an advantage for *Wolbachia* bacteria and increases the infected progeniture rate. Furthermore, *Wolbachia* go through an important recombination phenomenon between strains that affects various regions of the genome, such as surface proteins genes, housekeeping genes, prophage genes, and intergenic regions [21,22]. This could explain the extreme diversities of *Wolbachia* and their interactions with their hosts.

Epidemics of mosquito-borne diseases such as dengue, chikungunya, and Zika occur frequently in French Guiana, with a heavy burden on public health [23]. The difficulties in handling such arbovirus outbreaks in the territory, in relation to the high insecticide resistance of *Aedes aegypti*, testify to the need to develop new methods of control [24]. Ideally, it ought to be efficient, non-hazardous to human health, and environmentally friendly. Several *Wolbachia*-based methods have been developed and studied during the past few decades, with several field trials in different countries across the globe [25,26,27,28]. Two main strategies aim at either drastically reducing the population of *Aedes aegypti* mosquitoes using the incompatible insect technique [29]—a variant of the sterile insect technique, such as in Yucatan (Mexico) [30], or using blocking virus strategies with the aim of decreasing dengue virus transmission by vector population replacement [31,32], such as in Brazil [33], New Caledonia [34], and Indonesia [26]. Additionally, some countries have chosen to implement a combination of the incompatible insect technique (IIT, based on *Wolbachia* bacteria) and the sterile insect technique (SIT, based on radiation-sterilized insects) to decrease the dengue burden; they have demonstrated successful field trials and the feasibility of area-wide application of combined IIT-SIT for mosquito vector control [35]. 

In South America, replacement population programs have begun to be applied in some countries, such as Brazil [33,36]. Several studies [17,31,37] have shown the presence of *Wolbachia* in the mosquitoes of South America, but little is known about the diversity of the strains involved. When considering vector control based on these bacteria in French Guiana, it is important, as a baseline, to know the presence and distribution of *Wolbachia* in mosquito species, vectors or not. Moreover, the proof of *Wolbachia* already existing in human-biting mosquitoes in the French territory could play a role in population acceptance. This study aimed to highlight the diversity, prevalence, and distribution of *Wolbachia* bacteria in several mosquito species living in French Guiana.

## 2. Materials and Methods

The high diversity of mosquito species in French Guiana leads to some difficulties in their identification. A detailed workflow describing the entire process of mosquito species identification followed by *Wolbachia* infection verification and classification is presented in Figure 1.

### 2.1. Mosquito Collection and Identification

Mosquitoes were collected between 2018 and 2023 in eight different municipalities of French Guiana (Cayenne, Rémire-Montjoly, Matoury, Macouria, Régina, Roura, Saül, and Maripasoula) using four different sampling methods: BG-Sentinels traps, CDC-light traps, mosquito magnet traps, and immature stage collection (Figure 2). Conveniently, *Aedes aegypti* sampling was carried out using larval collections, particularly along the Maroni River (Maripasoula, Grand Santi, Apatou, and Saint Laurent) and the coastal area (Mana, Iracoubo, Sinnamary, Kourou, Tonate, Cayenne, and Matoury). The larvae were then reared to the adult stage for morphological identification and further analysis. After collection, mosquitoes were identified using morphological keys [38] when possible and dry stored at −20 °C before processing. In order to increase the range and number of tested species, some specimens were picked up from the institute’s biobank out of −20 °C storage freezers (collection up to 2009). Prior to DNA extraction, sterile pestle grinding was manually performed for each sample. Subsequently, DNA was individually extracted from each mosquito specimen using the QIAamp DNA tissue extraction kit (Qiagen, Hilden, Germany) following the manufacturer’s instructions.

### 2.2. Barcoding Culex Identification (COI)

All samples that could not be morphologically identified to the species level by using identification keys [38] were identified through DNA barcoding [39]. 

Molecular identification was performed using the mitochondrial cytochrome c oxidase I (COI) gene and the DNA reference library dedicated to the mosquitoes of French Guiana [40]. The HOT FIREPol DNA Polymerase Kit was used following the manufacturer’s instructions for PCR reaction mix (Solis BioDyne, Tartu, Estonia), with a hybridization temperature of 50 °C for COI primers (Table A1 and Table A2). The amplifications for the COI gene were verified by using QIAxcel Advanced (Hilden, Germany) on gene fragments with the QIAxcel DNA Screening Kit (Hilden, Germany) following the manufacturer’s instructions [41].

### 2.3. Wolbachia Detection Real-Time Quantitative PCR

A first screening of *Wolbachia* detection targeting the 16s rRNA gene in all samples was conducted using real-time PCR [42]. Briefly, the mosquitoes were screened individually using SuperScript™ III Platinum™ One-Step qRT-PCR Kit (Invitrogen, Waltham, MA, USA) and Wol_F and Wol_R primers [42] for real-time PCR with a Wol_P probe (Table A1 and Table A3). This initial step with high detection sensitivity [42] aimed at the identification of samples containing *Wolbachia*. The genotyping of *Wolbachia* from positive samples was then established when the threshold cycle value (Ct) was lower than 20.

### 2.4. Wolbachia 16S rRNA and WSP Typing

Two *Wolbachia* genes were sequenced out of the selected positive samples for the 16S rRNA gene and highly conserved [43,44] and WSP genes [22] (Table A1). PCR amplifications were performed using a HOT FIRE Pol DNA Polymerase kit (Solis BioDyne, Tartu, Estonia) following the previous PCR protocol with hybridization temperatures of 45 °C and 51 °C for 16S rRNA and WSP primers, respectively (Table A1 and Table A2). The amplification quality for 16S rRNA and WSP genes was checked by QIAxcel as previously described. Positive samples for both 16S rRNA and WSP genes were sequenced using the Sanger method (GENEWIZ, Leipzig, Germany). When multiple bands were obtained after WSP PCR, the amplicons were sequenced using the MinION Mk1C device (Oxford Nanopore Technologies, Oxford, UK). For MinION sequencing, the libraries were produced according to the base protocol from Oxford Nanopore [45] and the ARTIC nCoV-2019 sequencing protocol (https://www.protocols.io/view/ncov-2019-sequencing-protocol-v2-bdp7i5rn, accessed on 22 April 2024).

### 2.5. Wolbachia MLST Analysis

Five genes (CoxA, GatB, FbpA, FtsZ, and HcpA) [22,46] (Table A1) were amplified for MLST analysis [22], with slight modifications for the annealing temperatures (CoxA and FbpA: 53 °C, GatB: 54 °C, HcpA: 48 °C, and FtsZ: 58 °C). All positive samples for MLST gene detection were sequenced using the Sanger method (GENEWIZ, Leipzig, Germany). Sequences of each of the five genes were then compared to the MLST database (http://pubmlst.org, accessed on 5 January 2024). 

### 2.6. Nanopore Sequencing

After MinION sequencing, the raw reads were merged into a single fasta file using the artic guppyplex tool (https://github.com/artic-network/fieldbioinformatics, accessed on 22 April 2024) (version 1.2.1). 

The obtained fasta files were processed using NGSpeciesID (https://github.com/ksahlin/NGSpeciesID, accessed on 22 April 2024) [47] to obtain consensus before comparing them to the NCBInt database using the dchimer tool (https://github.com/stirera/d-chimer_v1, accessed on 22 April 2024). The fasta files were then mapped against a 575 bp long reference fasta file from the same host, obtained by Sanger sequencing, to obtain a visual output using Integrative genomic viewers (IGV).

### 2.7. Phylogenetic Analysis

All obtained sequences of the 16S rRNA, WSP, and COI genes were compared to the GenBank database [48] using Blastn (https://blast.ncbi.nlm.nih.gov/Blast.cgi, accessed on 15 March 2024). Moreover, an updated reference database of recently obtained Culex COI sequences accessible in the Barcode of Life Data System (BOLD) [49] was also used to identify mosquito species. 

The sequences obtained in this study were aligned using ClustalW (http://www.ebi.ac.uk/clustalw/, accessed on 22 April 2024) [50] with references from the GenBank, pubMLST, and *Culex* databases from Institut Pasteur de la Guyane. Maximum likelihood trees for phylogenetic analysis were built from Sanger sequences and the sequences obtained from MiniON sequencing, all conducted using MEGA X 10.2 software [51]. External sequences (retrieved from the NCBI database) from model organisms (*Dirofilaria*, *Drosophila*, *Aedes albopictus*, etc.) were included in the analysis for a better classification.

## 3. Results

A total of 507 individuals belonging to 15 genera and 44 species were collected (Appendix A). The most prominent genus (11 species) was *Culex*, followed by *Aedes* (8 species). A total of 24 species out of 44 were positive (54.5%) according to conventional PCR for the WSP gene (Figure 3), but a lower level of bacteria load was detected (ct < 25) via *Wolbachia* 16S rRNA RT-qPCR in 6 additional species (giving a total of 65.9% positive species) (Figure 3). The range of prevalence of *Wolbachia* extended from 5.6% for *Psorophora ferox* up to 100% for *Culex quinquefasciatus*. Importantly, all *Ae. aegypti* collected in 2019 and 2022 in French Guiana (Cayenne, Kourou, Grand Santi, Maripasoula, Apatou, Saint-Laurent, and Iracoubo) were found to be negative for *Wolbachia.*

For the *Mansonia titillans* species, no *Wolbachia* were found in the *Ma. titillans* Macouria specimens collected in 2022. However, the specimens collected in 2011 at Camp du Tigre, near Cayenne, were positive (8 out of 10; 80%) for *Wolbachia* infection. Three samples exhibited two similar profiles for the WSP gene. After Nanopore sequencing, the two obtained consensuses were aligned to *Wolbachia* WSP genes in each barcode or sample, following Blastn using the dchimer tool: a long sequence of 530 bp and a short sequence of 515 bp for each of the three barcodes. In both the bam (binary alignment map) files and consensuses, the short sequence has more depth of coverage than the longer one, which is in concordance with the gel density results. This is a strong indication of the presence of coinfection involving two *Wolbachia* strains.

### 3.1. Phylogenetic Analysis

#### 3.1.1. Maximum Likelihood Tree of Wolbachia Based on 16S rRNA Gene

The highly conserved 16S rRNA gene in the phylogenetic analysis (Figure 4) indicates that most of the *Wolbachia* collected from French Guiana mosquito species gather in the same cluster as supergroup B, such as wPip (MZ577347). The *Wolbachia* sequences from *Wyeomyia luteoventralis* and *Sabethes quasicyaneus* gather tightly with *Trichoprosopon digitatum* in a peripherical cluster of this large supergroup B. Additionally, the strains found in *Coquillettidia venezuelensis* and *Uranotaenia leucoptera* cluster with the *Wolbachia* strains from supergroup A, such as wMel (DQ412083).

#### 3.1.2. Maximum Likelihood Tree of Wolbachia Based on the WSP Gene

The WSP gene, with a higher evolution rate, provides a slightly different phylogeny (Figure 5). A total of 25 strains were characterized and compared to the GenBank database sequences. As with 16S rRNA, *Cq. venezuelensis* and *Ur. leucoptera* are related to *Glossina morsitans* (JF494897) and are clustered together within supergroup A, such as with *Ae. albopictus* (AY462864). Even if *Culex* species cluster differently, they can be found only in supergroup B. Interestingly, *Wolbachia* strains coinfecting three *Ma. titillans* (EC065, EC066, and EC067) specimens present two different positions in the tree in the same supergroup B, while the *Ma. titillans* specimen (EC070) presenting one unique *Wolbachia* strain clustered with one of the two *Wolbachia* strains infecting the three *Ma. titillans* (ii) specimens.

#### 3.1.3. Maximum Likelihood Tree (Parenting Links) for MLST Analysis

All of the loci used for MLST analysis were not amplified in every species. Consequently, only 12 strains were fully characterized by MLST, and these were compared to nine fully characterized strains retrieved from the PubMLST database. None of the concatenated five loci matches perfectly with the sequences already present in the PubMLST database and, therefore, represent original sequences. For example, this is the case for *Cx. Quinquefasciatus*, which was collected during our study, when compared to *Cx. Quinquefasciatus*, which was collected in the USA (wPip iso 1808). Most of the strains from supergroup B highlighted in this study (Figure 6) likely descend from a recent common ancestor with a major clade with nine species. Three strains cluster differently. Globally, the distribution of the sequences obtained in this study is still consistent with the distribution obtained using the phylogenetic analysis of the WSP gene, with most of the species attached to supergroup B, and *Ur. leucoptera* and *Cq. venezuelensis* are both closer to supergroup A.

## 4. Discussion

### 4.1. Wolbachia Detection

Mosquitoes were collected using different traps, maximizing the diversity. A total of 44 species were tested out of the 242 [11] (or 18.6%) mosquito species recorded in French Guiana. Fieldwork was carried out mainly on the coast, in the populated area of the territory with only a few locations inland, in Saül or Maroni River villages.

A few species within the *Aedes* genus were found to be naturally infected by the bacteria (*Ae. scapularis* and *Aedes serratus*). No *Ae. fluviatilis* was found to be naturally infected (PCR and RT-qPCR) despite this species having been previously found to be infected with the bacteria in other geographical locations [17]. Interestingly, *Ae. aegypti,* the major vector of dengue, is not naturally infected by *Wolbachia* in French Guiana in any of the tested collection sites. Natural *Wolbachia* infection is rarely found in *Ae. aegypti*. Currently, studies on the natural purported infection of *Ae. aegypti* with *Wolbachia* have only been reported in the USA [54,55,56], Malaysia [57], Thailand [58], India [59], the Philippines [44], and Panama [60]. Although not naturally present in *Ae. aegypti*, the wMel strain was stably introduced into this mosquito in 2009 [61] and has been shown to reduce the transmission potential of dengue, Zika, and chikungunya. *Aedes aegypti* carrying the wMel [62] or wAlbB [63] strains of *Wolbachia* have the potential to reduce dengue transmission through decreased mosquito vector competence, and there is already good evidence that both strains are having such impacts in *Wolbachia*-invaded release areas. 

Some studies suggest several requirements to confirm a *Wolbachia* natural infection [64]. Accordingly, this study used seven different genes: 16S rRNA, WSP, CoxA, GatB, HcpA, FtsZ, and FbpA. The 16S rRNA gene marker is known to perform well, and the WSP gene marker completes this type of molecular screening. For the primers used in the MLST analysis, it was difficult to obtain all markers completely for all specimens, and obtaining an individual that was positive for all *Wolbachia* detection markers was rare. On the other hand, we are confident that this complementary approach would minimize the false negative bias, giving us a comfortable view of the detection rate and prevalence values.

The difficulties in detecting the bacteria in species with low prevalence could be due to the low density of the bacteria within individuals [65]. A total of 29 mosquito species were found to be positive (16S rRNA PCR) for *Wolbachia* natural infection. The infection was confirmed by a second PCR targeting WSP. The samples with low (ct > 20) CT values for 16S rRNA qPCR were negative for most of them according to WSP PCR. With this approach monitoring the WSP signal only, the prevalence of *Wolbachia* within a species varied from 5.3% (for *Ae. scapularis)* up to 100% for *Cx. Quinquefasciatus.* A total of 15 mosquito species were infected with a *Wolbachia* prevalence of higher than 50%. The low prevalence of *Wolbachia* in some species, according to two molecular markers, can be explained by an imperfect maternal transmission added to the absence of paternal transmission and/or random mating [66,67]. The higher the infection rate is, the higher the reproductive advantage [67,68] and the higher the chance for *Wolbachia* to spread into the population. 

### 4.2. Prevalence Difference

Among the positive species, *Coquillettidia albicosta, Culex portesi,* and *Aedeomyia squamipennis* showed a high prevalence, ranging from 75 to 86%, with a low Ct for *Wolbachia* detection.

*Aedes scapularis* has been recently reported to carry *Wolbachia* [37] at comparably low levels in Brazil (12.5%) and French Guiana (5.2%). *Coquillettidia venezuelensis* was found with a higher prevalence (63.2%) in French Guiana than in Brazil (20%) [37]. *Culex quinquefasciatus* has been known to be a *Wolbachia* carrier for a long time and is well known for its vector potential for pathogens such as filariasis and West Nile virus. To our knowledge, this study is the first to report *Wolbachia* infections in several species: *Ur. leucoptera*, *Cq. albicosta*, *Wy. luteoventralis*, *Sa. quasicyaneus*, *Culex idottus*, *Culex rabanicolus,* and *Culex eastor*. Additionally, the *Sabethes* and *Wyeomyia* genera, despite already being screened in the past [69], have been found to be positive for the first time. No *Wolbachia* infection has been found in the Yellow Fever vector *Haemagogus janthinomys,* but the limited availability of samples precludes definitive conclusions. 

### 4.3. Typing and Classification

The 16SrRNA gene, historically used to detect *Wolbachia* [70], gives us the first phylogenetic representation relationship between *Wolbachia* and its host. However, WSP gene sequencing is frequently used to type *Wolbachia* strains, giving a good discrimination of strain diversity [42]. Moreover, the MLST analyses, which have recently been developed to increase precision, confirm the detection of new and original *Wolbachia* strains identified in French Guiana.

Here (Figure 5), the WSP phylogenetic tree shows that supergroup B is strongly represented by a large part of *Wolbachia* strains infecting Guianese mosquito species. Only three species are clustered in supergroup A: for the first time, the *Uranotaenia* genus was found to carry the *Wolbachia* strain from supergroup A [65]; the same thing is observed for *Cq. venezuelensis* [37]. The prevalence of the bacteria from a species with a prevalence higher than 50% to another species has the potential to significantly impact the evolutionary dynamics of the recipient species, potentially leading to adaptations that improve its survival and reproductive success through the *Wolbachia*–host symbiotic relationship.

### 4.4. Diversity by Host Species

Even with the limited number of *Wolbachia* strains analyzed in this study, some patterns regarding the diversity and the transmission of the bacteria are revealed. 

The coherent phylogenetic proximity according to the three classifications between *Cq. venezuelensis* and *Ur. leucoptera* could demonstrate a recent horizontal transfer of *Wolbachia* between different genera. The same pattern is found within the WSP and 16S phylogenetic trees, which set *Sa. quasicyaneus* and *Wy. luteoventralis* together in supergroup B. These examples could support the recent horizontal transfer of bacteria. The aquatic stage could be a hypothetical spot of exchange. Indeed, *Sa*. *quasicyaneus* and *Wy. luteoventralis* are the only species of this study for which the larvae are both found in small breeding sites located in leaf hollows. Otherwise, horizontal transfers remain rare because *Wolbachia* is obligately intracellular. Therefore, introgressive transfer could be another hypothesis for explaining this closely related *Wolbachia* strain [71]. In order to confirm this event, more analysis comparing the genome of *Wolbachia* strains and the mtDNA of the host genome should be performed to highlight the divergence time, indicating if *Wolbachia* are transferred via introgression or horizontal transfer. 

Of note, three species of the *Culex* (*Melanoconion*) genus, *Cx. idottus, Cx. Eastor*, and *Cx. rabanicolus*, were collected in the Cacao village on the same day and within the same trap (MosquitoMagnet), evidencing a shared habitat as adults. However, these three phylogenetically related mosquito species carry three different *Wolbachia* strains. *Culex idottus* and *Cx. rabanicolus* carry the same *Wolbachia* strain as *Cx. Portesi*, while *Cx. eastor* specimens have two different and distant strains. 

### 4.5. Diversity within Individuals of the Same Species

*Wolbachia* have already been recorded in *Ma. titillans* from Cuba (ON550483) [72] and Brazil [17]. However, the *Wolbachia* strains from *Ma. titillans* in our study do not cluster with any of the *Wolbachia* strains from *Ma. titillans* from Cuba and *Mansonia uniformis* from India [73]. Nanopore MinION sequencing revealed two different strains of *Wolbachia* in three specimens of *Ma. titillans*, evidencing dual infection by two different *Wolbachia* strains in the same supergroup B. This example adds a new case of dual natural infection of *Wolbachia* in mosquitoes, such as the case reported in *Ae. albopictus* by Dutton et al. (2004) [74].

## 5. Conclusions

This study contributes to the knowledge about the diversity of mosquitoes associated with *Wolbachia* bacteria in French Guiana, introducing new strains and discovering new host species. Communicating this information to local populations is crucial, as future strategies for protecting humans and domestic animals from mosquito-borne diseases might involve alternatives to chemical insecticides, such as *Wolbachia*-based control methods. These approaches require a deep understanding of *Wolbachia*–mosquito associations and the acceptance of local communities. The diversity of Wolbachia strain combinations that infect natural mosquito populations presents phenotypes to test for in terms of reducing or altering pathogen transmission in mosquito populations. The potential of new *Wolbachia* strains to block viruses and manipulate host reproduction, density, tissue tropism, and sensitivity to temperature, as well as the associated fitness costs in specific host–microbe relationships, are important areas for future research.

## Figures and Tables

**Figure 1 microorganisms-12-01994-f001:**
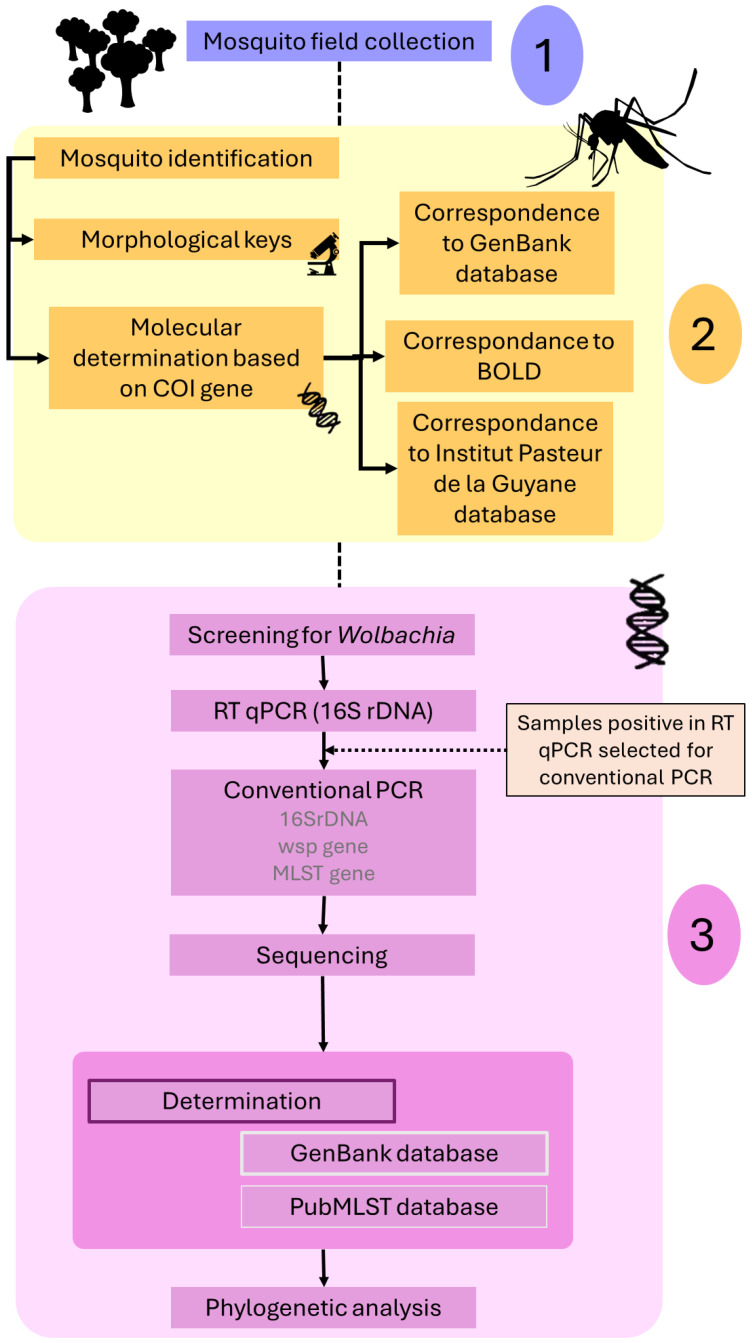
Workflow of the methodological approach to mosquito species and *Wolbachia* identification.

**Figure 2 microorganisms-12-01994-f002:**
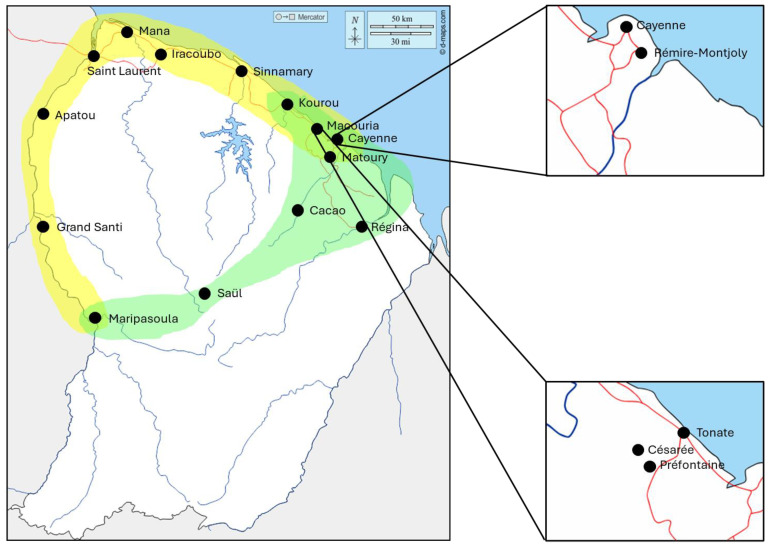
Location of collection points of mosquitoes in French Guiana. In green, mosquitoes were collected in light traps in Macouria (Tonate, Préfontaine, and Césarée), Cayenne (center of Cayenne and Rémire-Montjoly), Kourou, and Stoupan. BG-Sentinel traps were used in Saül and Macouria. Mosquito magnets were used in Cacao, Macouria, and Régina. Immature stages were collected along the Maroni River (Maripasoula Grand Santi, Apatou, and Saint-Laurent) and along the coast (Mana, Iracoubo, Sinnamary, Kourou, and Cayenne). In yellow, *Aedes aegypti* larval collections. In green, adult mosquito species collection [d-maps.com].

**Figure 3 microorganisms-12-01994-f003:**
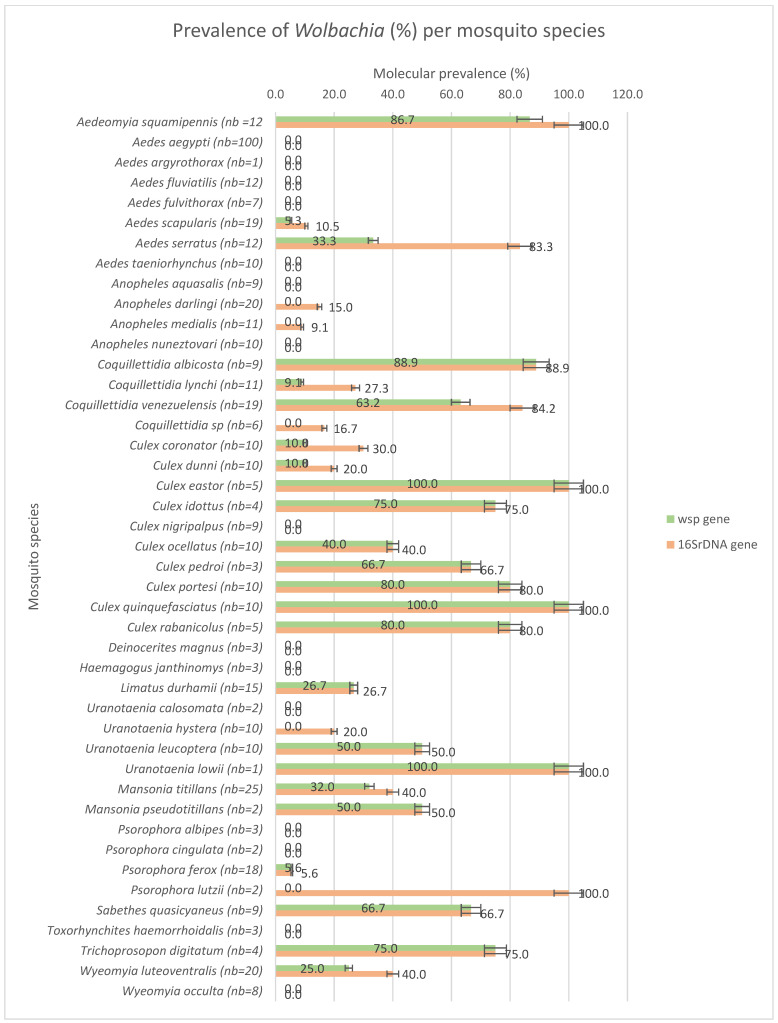
Molecular prevalence of *Wolbachia* in mosquito species (number of individuals screened is indicated in brackets next to the mosquito species name) according to 16S rRNA and WSP genes.

**Figure 4 microorganisms-12-01994-f004:**
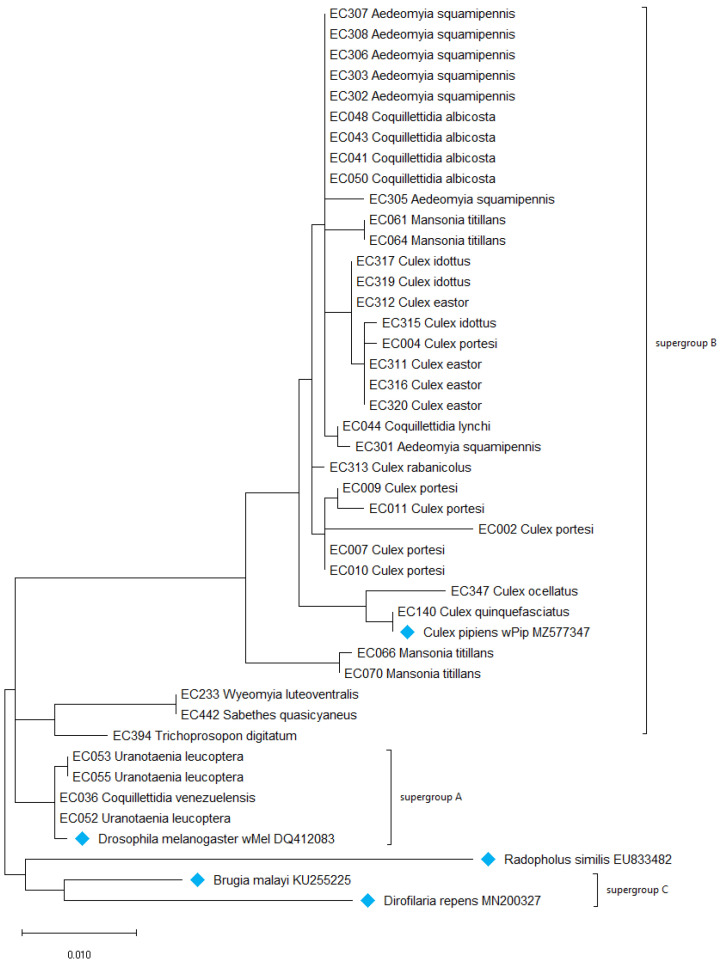
The evolutionary history of the 16S rRNA gene was inferred by using the maximum likelihood method and the Tamura–Nei model [52]. The samples are identified with the name of their host. The tree with the highest log likelihood (−3488.39) is shown. Bootstrap values are shown next to the branches. Initial tree(s) for the heuristic search were obtained automatically by applying the Neighbor Joining and BioNJ algorithms to a matrix of pairwise distances estimated using the maximum composite likelihood (MCL) approach and then selecting the topology with superior log likelihood value. The tree is drawn to scale, with the branch lengths measured in the number of substitutions per site. This analysis involved 43 nucleotide sequences. The codon positions included were 1st + 2nd + 3rd + noncoding. There are 850 positions in the final dataset. The evolutionary analyses were conducted in MEGA X [53]. The blue labels show sequences retrieved from the NCBI database.

**Figure 5 microorganisms-12-01994-f005:**
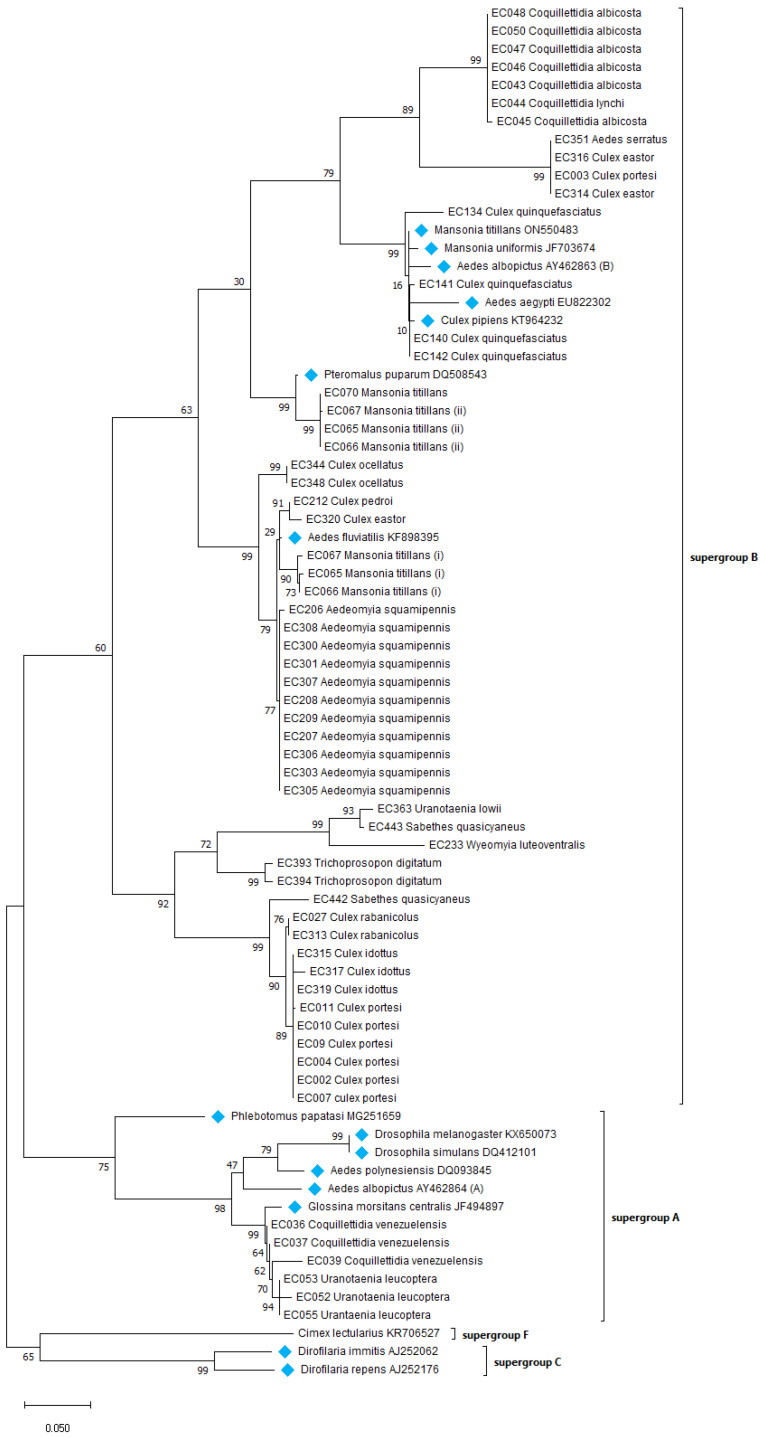
The evolutionary history of the WSP gene was inferred by using the maximum likelihood method and Tamura–Nei model and shows the name of the host where *Wolbachia* were detected. The tree with the highest log likelihood (−5397.55) is shown. Bootstrap values are shown next to the branches. Initial tree(s) for the heuristic search were obtained automatically by applying the Neighbor Joining and BioNJ algorithms to a matrix of pairwise distances estimated using the maximum composite likelihood (MCL) approach and then selecting the topology with the superior log likelihood value. The tree is drawn to scale, with the branch lengths measured in the number of substitutions per site. This analysis involved 75 nucleotide sequences. The codon positions included were 1st + 2nd + 3rd + noncoding. There are 688 positions in the final dataset. The evolutionary analyses were conducted in MEGA X. The blue labels show the sequences retrieved from the NCBI database.

**Figure 6 microorganisms-12-01994-f006:**
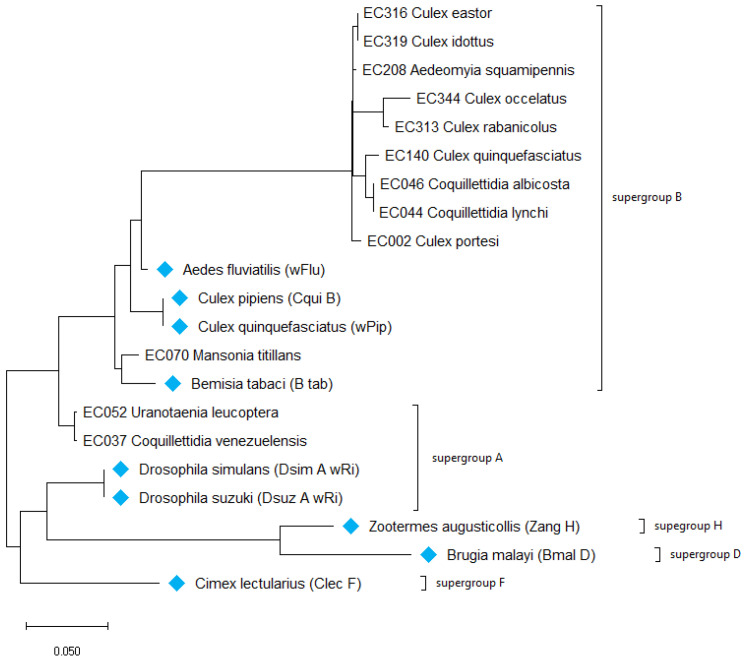
The evolutionary history of MLST loci was inferred by using the maximum likelihood method, and the Tamura–Nei model shows the name of the host where *Wolbachia* were detected. The tree with the highest log likelihood (−9930.75) is shown. Bootstrap values are shown next to the branches. Initial tree(s) for the heuristic search were obtained automatically by applying the Neighbor Joining and BioNJ algorithms to a matrix of pairwise distances estimated using the maximum composite likelihood (MCL) approach and then selecting the topology with the superior log likelihood value. The tree is drawn to scale, with the branch lengths measured in the number of substitutions per site. This analysis involved 21 nucleotide sequences. There is a total of 1984 positions in the final dataset. The evolutionary analyses were conducted in MEGA X. The blue labels show the sequences retrieved from the PubMLST database.

## Data Availability

The datasets generated and/or analyzed during the current study are available in the GenBank database repository. Please find the WSP (Appendix A) and 16SrRNA (Appendix A) gene sequences and their accession numbers below. The sequences generated and used in this work for MLST analysis are available from the corresponding authors.

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
