# Peer review of "Wolbachia Natural Infection of Mosquitoes in French Guiana: Prevalence, Distribution, and Genotyping"

_microorganisms, 2024, doi:10.3390/microorganisms12101994_

Round 1

Reviewer 1 Report

Comments and Suggestions for Authors

Mosquito Collection:

·         Were only Aedes aegypti mosquitoes collected through larval collection? If so, why? Are larvae collected from the same location and time likely to be from the same adult female? How did the authors ensure that this collection method did not reduce the variability of the samples?

·         How many samples were collected from the field? How many were obtained from the institute’s biobank, and what species were included in the biobank?

·         Line 107: Was only Culex barcoded?

Figure 3: How did the authors define Wolbachia-positive and -negative samples? Was this determination based on qPCR or sequencing? In the methods section, only one gene was screened for by qPCR, whereas two genes were used for sequencing. So please clarify.

Line 296: The authors mention "low Ct" but do not provide information about the Ct results in the manuscript.

Line 334: Again, the authors did not provide information about the time and location distribution of the collected samples.

Author Response

Reviewer 1 :

Mosquito Collection:

  • Were only Aedes aegypti mosquitoes collected through larval collection? If so, why? Are larvae collected from the same location and time likely to be from the same adult female? How did the authors ensure that this collection method did not reduce the variability of the samples?

Yes, only Aedes aegypti were collected at larval stage. But they were collected in several localization and in different breeding site per localization.

  • How many samples were collected from the field? How many were obtained from the institute’s biobank, and what species were included in the biobank?

401 individuals were collected from the field. 107 individuals were retrieved from the biobank belonging to 20 different species. No other species were added to the biobank, collected field mosquitoes during the project were already in the biobank.

  • Line 107: Was only Culex barcoded?

No, all other species were barcoded, but Culex species were identified from dataset previously generated in Institut Pasteur de la Guyane. Sequences of Culex species are available on BOLD database.

 Figure 3: How did the authors define Wolbachia-positive and -negative samples? Was this determination based on qPCR or sequencing? In the methods section, only one gene was screened for by qPCR, whereas two genes were used for sequencing. So please clarify.

Positive samples in qPCR (ct<25) were then screened in PCR. Positive sample to PCR were then sequence in SANGER

 Line 296: The authors mention "low Ct" but do not provide information about the Ct results in the manuscript.

 Line 334: Again, the authors did not provide information about the time and location distribution of the collected samples.

Fig2, the map gives us all the collection point of mosquito we collected.

Reviewer 2 Report

Comments and Suggestions for Authors

This is an important study about Wolbachia infection in French Guiana mosquitoes, the authors clarified the Wolbachia strains, distribution, and prevalence in mosquitoes of French Guiana, which would be a great fundament to further use Wolbachia to block virus in French Guiana. However, many works should be done before publication. Following notes are my comment:

Major concerns:

1.             The authors sequenced Wolbachia genes such as WSP or MLST genes as well as mosquitoes COI gene, while no comparison between Wolbachia genes and host genes was present and discussed. In many research, there is a strong co-evolution relationship between host and symbiont. The authors could compare the relationship between Wolbachia and host by building phylogenetic analysis of the two mutualistic factors.

2.             In phylogenetic analysis, ITS genes of animals are also important to analyze the relationship between Wolbachia and host mosquitoes. If possible, authors may be better to provide the ITS sequencing results and phylogenetic analysis.

3.             The authors could combine the 3 Wolbachia genes (16S rRNA gene, WSP and MLST genes) to build a more reliable phylogenetic analysis of Wolbachia.

4.             In introduction, authors should add more information about Wolbachia control virus in mosquitoes, such as works in Australia and Southest Aisa by A.A. Hoffman and Zhiyong Xi.

5.             In results part, no need to describe the method of this experiment, such as in line 180-182, the description of method should be replaced to material part.

6.          In the related research, the article entitled “Rapid global spread of wRi-like Wolbachia across multiple Drosophila” is an important work, which the authors should fully understand and discussed in this paper.

Minor concerns:

1.             The first two paragraphs of introduction should be combined to one paragraph.

2.             In paragraph line 38-52, the popular name and scientific name of different species are mixed-use, which use could better to be consistent.

3.             In line 22, “Wolbachia” should be italic.

4.             In line, “diversity” should be “diversities”.

5.             In line 29, there should be a space between “1924” and superscript “4”.

6.             Line 52, “host” should be “hosts”.

7.             In line 61, the underline should be deleted, and the space between “Yucatan” should be deleted.

8.             Line 65-74 should be one paragraph.

9.             Line 163, the mixed-use of popular name and scientific name.

10.          Line 193, “wPip” should be “wPip”.

11.          Line 198, “wMel” should be “wMel”, and all similar errors should correct in following parts.

12.          Some errors in reference, such as reference 19, the journal name is not abbreviation. Reference 26 the journal name abbreviation is not correct. Reference 69 the capital letters should be corrected.

13.          In all references, the scientific names are not accurately italic.

Comments on the Quality of English Language

Minor concerns:

Author Response

Reviewer 2

Comments and Suggestions for Authors

This is an important study about Wolbachia infection in French Guiana mosquitoes, the authors clarified the Wolbachia strains, distribution, and prevalence in mosquitoes of French Guiana, which would be a great fundament to further use Wolbachia to block virus in French Guiana. However, many works should be done before publication. Following notes are my comment:

 Major concerns:

  1. The authors sequenced Wolbachia genes such as WSP or MLST genes as well as mosquitoes COI gene, while no comparison between Wolbachia genes and host genes was present and discussed. In many research, there is a strong co-evolution relationship between host and symbiont. The authors could compare the relationship between Wolbachia and host by building phylogenetic analysis of the two mutualistic factors.

We observed and underlined several cases where the phylogeny between the hosts and the Wolbachia symbiont does not fit: as examples Mansonia titillans, Coquilletidia venzuelensis or Uranotaenia leucoptera

  1. In phylogenetic analysis, ITS genes of animals are also important to analyze the relationship between Wolbachia and host mosquitoes. If possible, authors may be better to provide the ITS sequencing results and phylogenetic analysis.

We agree for the interests of IRTS marker, especially for Anopheline species.  Indeed, our group has already published RFLP tool after this marker (Venezegho et al, 2022). However, despite initial promising works (Navarro and Weaver, 2004) and considering the wide diversity within the Amazonian Culex, CO1 has revealed more potential than ITS. We disagree with the allegedly added value of ITS in this work.

  1. The authors could combine the 3 Wolbachia genes (16S rRNA gene, WSP and MLST genes) to build a more reliable phylogenetic analysis of Wolbachia.

It could be very interesting to combined 3 genes together, but we couldn’t amplified and sequence all genes for all species. For example, we have many species with wsp genes amplified and sequenced but couldn’t do the same for the 5 MLST genes and analysis.

In introduction, authors should add more information about Wolbachia control virus in mosquitoes, such as works in Australia and Southest Aisa by A.A. Hoffman and Zhiyong Xi.

  1. In results part, no need to describe the method of this experiment, such as in line 180-182, the description of method should be replaced to material part.
  2. In the related research, the article entitled “Rapid global spread of wRi-like Wolbachia across multiple Drosophila” is an important work, which the authors should fully understand and discussed in this paper.

Round 2

Reviewer 2 Report

Comments and Suggestions for Authors

The authors fully correct the manuscript, while the reference should be modified, such as incorrect capital letters of reference 58 and 72, some needless DOI number in references. The format of line 70-72 should be modified.